# Lurasidone versus Quetiapine for Cognitive Impairments in Young Patients with Bipolar Depression: A Randomized, Controlled Study

**DOI:** 10.3390/ph15111403

**Published:** 2022-11-14

**Authors:** Xiangyuan Diao, Dan Luo, Dandan Wang, Jianbo Lai, Qunxiao Li, Peifen Zhang, Huimin Huang, Lingling Wu, Shaojia Lu, Shaohua Hu

**Affiliations:** 1Department of Psychiatry, The First Affiliated Hospital, Zhejiang University School of Medicine, Hangzhou 310003, China; 2Department of Psychiatry, First Affiliated Hospital, Jiaxing University, Jiaxing 314001, China; 3Department of Psychosomatic, The Third People′s Hospital of Jiashan County, Jiaxing 314100, China; 4Department of Psychiatry, Hangzhou Fuyang Third People’s Hospital, Hangzhou 311402, China; 5Brain Research Institute, Zhejiang University, Hangzhou 310003, China; 6Zhejiang Engineering Center for Mathematical Mental Health, Hangzhou 310003, China; 7NHC and CAMS Key Laboratory of Medical Neurobiology, School of Brain Science and Brain Medicine, MOE Frontier Science Center for Brain Science and Brain-Machine Integration, Zhejiang University School of Medicine, Hangzhou 310003, China

**Keywords:** quetiapine, lurasidone, bipolar disorder, cognition, adolescent

## Abstract

The clinical efficacy of lurasidone and quetiapine, two commonly prescribed atypical antipsychotics for bipolar depression, has been inadequately studied in young patients. In this randomized and controlled study, we aimed to compare the effects of these two drugs on cognitive function, emotional status, and metabolic profiles in children and adolescents with bipolar depression. We recruited young participants (aged 10–17 years old) with a *DSM-5* diagnosis of bipolar disorder during a depressive episode, who were then randomly assigned to two groups and treated with flexible doses of lurasidone (60 to 120 mg/day) or quetiapine (300 to 600 mg/day) for consecutive 8 weeks, respectively. All the participants were clinically evaluated on cognitive function using the THINC-it instrument at baseline and week 8, and emotional status was assessed at baseline and the end of week 2, 4, and 8. Additionally, the changes in weight and serum metabolic profiles (triglyceride, cholesterol, and fasting blood glucose) during the trial were also analyzed. In results, a total of 71 patients were randomly assigned to the lurasidone group (*n* = 35) or the quetiapine group (*n* = 36), of which 31 patients completed the whole treatment course. After an 8-week follow-up, participants in the lurasidone group showed better performance in the Symbol Check Reaction and Accuracy Tests, when compared to those in the quetiapine group. No inter-group difference was observed in the depression scores, response rate, or remission rate throughout the trial. In addition, there was no significant difference in serum metabolic profiles between the lurasidone group and the quetiapine group, including triglyceride level, cholesterol level, and fasting blood glucose level. However, the quetiapine group presented a more apparent change in body weight than the lurasidone group. In conclusion, the present study provided preliminary evidence that quetiapine and lurasidone had an equivalent anti-depressive effect, and lurasidone appeared to be superior to quetiapine in improving the cognitive function of young patients with bipolar depression.

## 1. Introduction

Bipolar disorder (BD) is a serious and highly disabling mental illness that affects at least 1% of the global population. Patients with BD always have an early age of onset, and as many as 60% of first episodes occurred in childhood or adolescence [1,2,3]. Childhood-onset of BD was often associated with longer treatment delays, more severe depressive episodes, an increased risk of suicide attempts [4], a higher prevalence of concurrent anxiety and substance abuse, as well as a poorer prognosis, compared with those with a late onset [5]. 

It is noteworthy that cognitive impairment is an intractable core feature of BD [6]. Previous studies indicated that cognitive function decreased significantly in patients with BD during acute episodes and might persist during euthymic periods [7]. Moreover, the cognitive impairment in BD might be more severe than that in unipolar disorder [8]. There is also mounting evidence that cognitive impairment may contribute to functional disability in patients with BD [9]. In the majority of studies on patients with BD, poorer cognitive function was associated with worse functional outcomes [10]. Therefore, the improvement of cognitive functioning should be an important treatment priority in individuals with BD. Worse still, patients with BD have significantly higher co-occurring rates of obesity, dyslipidemia, insulin resistance, and metabolic syndrome than those without BD [11]. The 2020 American Heart Association report showed that young adults susceptible to BD had an increased risk of atherosclerosis and early cardiovascular disease [12]. In the general population, an increase in body weight by one kilogram (kg) was equivalent to an increase by 3.1% in the risk of developing cardiovascular disease [13]; and an increase in BMI by one kg/m² will result in an increased risk of heart failure by 5–7% [14], and type 2 diabetes by 8.4% [15].

In the whole course of BD, depressive episodes occupied most of the stages [16]. Currently, the United States Food and Drug Administration (FDA) approved several drugs for the treatment of adult bipolar depression in the acute phase, including the olanzapine-fluoxetine combination, quetiapine, and lurasidone [17]. Among these treatment alternatives, lurasidone has demonstrated efficacy in the treatment of children and adolescents with bipolar depression and has been approved by the FDA as a monotherapy for bipolar depression in young patients aged 10–17 years [18]. Lurasidone, a full antagonist at dopamine D_2_ and serotonin (or 5-hydroxytryptophan [5-HT]) 2A (5-HT_2A_) receptor, also has a high affinity for serotonin 5-HT_7_ receptor and acts as a partial agonist at 5-HT_1A_ receptor. The 5-HT_7_ receptor is currently known to be involved in cognitive functions such as learning and memory [19]. In addition, lurasidone has been shown to reverse scopolamine and MK-801-induced impairment in learning and memory in preclinical models, suggesting its potential effects for protecting cognitive cognition [20,21]. Furthermore, previous clinical trials have shown preliminary evidence for the procognitive effects of lurasidone in patients with schizophrenia [22,23]. However, evidence on the effects of lurasidone on cognitive function in young patients with BD is still scarce.

Therefore, the first aim of the current study was to assess whether lurasidone was superior to quetiapine in the improvement of cognitive function in children and adolescents with bipolar depression; and the second aim was to compare the anti-depressive efficacy and metabolic side effects between these two atypical antipsychotics.

## 2. Results

### 2.1. Subject Characteristics

Of the 85 patients screened, 61 were found to be eligible for the study and were randomly assigned to the 8-week treatment (lurasidone, *n* = 29; quetiapine, *n* = 32). Of them, complete data were obtained from 61 patients at the first follow-up (2 weeks), 50 patients at the second follow-up (4 weeks), and 43 patients at the third follow-up (8 weeks). At the end of the 8th week, a total of 31 cases completed the cognitive function assessment (lurasidone, *n* = 17; quetiapine, *n* = 14). At baseline, there were no differences in age, gender, education level, weight, course of disease, height, tobacco, and alcohol use, or family history (*p* > 0.05) (Table 1). Patients in the lurasidone group were more likely to present psychotic symptoms (*p* = 0.045).

### 2.2. Primary Outcome

#### Cognitive Function

The primary outcome measure was to compare the difference in cognitive function at 8-week follow-up between the lurasidone and quetiapine groups. There was no difference in cognitive function at baseline between the two groups using the THINC-it tool (*p* > 0.05). According to covariance analysis, the change in the Symbol Check reaction time test and accuracy test before and after treatment in the lurasidone group was greater than that in the quetiapine group (LSM: 0.07, 95% CI: 0.02–0.11, *p* = 0.008; LSM: 0.16, 95% CI: 0.04–0.28, *p* = 0.012, respectively). However, no difference was found in the other four tests between the two groups (*p* > 0.05) (Table 2).

### 2.3. Secondary Outcome

#### 2.3.1. Depression Severity

According to the GEE analysis, there was no difference in the severity of depression between the two groups after intervention (Wald = 1.94, *p* = 0.164), as well as any evaluation point (Week 2: LSM: −1.43, 95% CI: −3.89–1.04, *p* = 0.256; Week 4: LSM: −1.87, 95% CI: −4.66–0.92, *p* = 0.189; Week 8: LSM: −1.66, 95% CI: −5.36–2.05, *p* = 0.381) (Figure 1A).

#### 2.3.2. Clinical Response Rate

There was no difference between the two groups regarding clinical response rate at 2, 4, or 8 weeks (lurasidone group: 17.6%, 58.8%,5 6.3%, respectively; quetiapine group: 28.6%, 71.4%, 64.3%, respectively) (Week2: LSM: 0.11, 95% CI: −0.19–0.41, *p* = 0.473; Week 4: LSM: 0.13, 95% CI: −0.21–0.46, *p* = 0.458; Week 8: LSM: 0.09, 95% CI: −0.26–0.44, *p* = 0.600) (Figure 1B). 

#### 2.3.3. Clinical Remission Rate

There was no difference between the two groups in terms of clinical remission rate at 2, 4, or 8 weeks (lurasidone group: 0.0%, 29.4%, 12.5%, respectively; quetiapine group: 7.1%, 28.6%, 35.7%, respectively) (Week 2: LSM: 0.07, 95% CI: −0.06–0.21, *p* = 0.299; Week 4: LSM: −0.01, 95% CI: −0.33–0.31, *p* = 0.959; Week 8: LSM: 0.24, 95% CI: −0.06–0.53, *p* = 0.120) (Figure 1C).

#### 2.3.4. Rate of Change in Body Weight

According to the GEE analysis, there were differences between the two groups after intervention (Wald = 3.90, *p* = 0.048). Compared with the lurasidone group, the rate of change in body weight of the subjects in the quetiapine group was greater at week 2 (LSM: 0.03, 95% CI: 0.00 –0.06, *p* = 0.037) and week 4 (LSM: 0.05, 95% CI: 0.00–0.09, *p* = 0.030), but not at week 8 (LSM: 0.03, 95% CI: −0.01–0.08, *p* = 0.140) (Figure 2A).

#### 2.3.5. Triglyceride Level

According to the GEE analysis, there was no difference between the two groups after intervention (Wald = 1.61, *p* = 0.205). Compared with the lurasidone group, the triglyceride level of the subjects in the quetiapine group showed no difference at baseline (t = 0.23, 95% CI: −0.44–0.55, *p* = 0.821), and week 4 (LSM: 0.11, 95% CI: −0.49–0.70, *p* = 0.725), but there was difference at week 8 (LSM: −0.53, 95% CI: −0.89–−0.17, *p* = 0.004) (Figure 2B).

#### 2.3.6. Cholesterol Level

According to the GEE analysis, there was no difference between the two groups after intervention (Wald = 0.35, *p* = 0.556). Compared with the lurasidone group, the cholesterol level of the subjects in the quetiapine group was not different at baseline (*t* = 0.30, 95% CI: −0.58–0.43, *p* = 0.765), week 4 (LSM: −0.26, 95% CI: −0.82–0.29, *p* = 0.349), and week 8 (LSM: −0.03, 95% CI: −0.59–0.54, *p* = 0.927) (Figure 2C).

#### 2.3.7. Fasting Blood Glucose Level

According to the GEE analysis, there was no difference between the two groups after intervention (Wald = 0.61, *p* = 0.434). Compared with the lurasidone group, the fasting blood glucose level of the subjects in the quetiapine group was not different at baseline (*t* = 0.98, 95% CI: −0.18–0.49, *p* = 0.339), week 4 (LSM: −0.14, 95% CI: −0.42–0.14, *p* = 0.328), and week 8 (LSM: −0.11, 95% CI: −0.63–0.41, *p* = 0.680) (Figure 2D).

## 3. Discussion

This study compared the effects of lurasidone and quetiapine on improving cognitive function, as well as depressive symptoms and metabolic profiles, in children and adolescents with bipolar depression under a randomized controlled design. Although no difference was found between the two groups on most tests of THINC-it, patients in the lurasidone group performed better in the Symbol Check reaction and accuracy tests than those in the quetiapine group, which is mainly relevant to the attention, memory, and reaction speed of the subjects.

It was previously demonstrated that adolescent patients with BD have a significantly poorer neurocognitive function in domains of attentional set-shifting, sustained attention, visuospatial memory, verbal declarative memory, processing speed, and executive functioning [24,25,26,27,28,29]. Biederman and collaborators evaluated 240 children and considered the executive functioning etiologically relevant to the behavioral symptoms of pediatric BD, while verbal learning and processing speed were considered as after-effects of the clinical condition [30].

In previous studies, the potential cognition-improving effects of lurasidone observed have been documented [20,21]. Although the exact mechanisms underlying the effect of lurasidone on cognition are unknown, its high affinity for 5-HT_7_ receptors might be an important contributor. Previous studies have found that short-term administration of lurasidone reversed the cognitive impairment induced by subchronic administration of phencyclidine (PCP), an NMDAR noncompetitive antagonist. Pharmacological data also suggest that the effect of lurasidone on NMDARs is likely to be through a mechanism involving 5-HT_7_ receptor antagonism. The 5-HT_7_ receptor is enriched in brain regions such as the limbic system, hippocampus, amygdala, and prefrontal cortex and mediates complex cognitive processes. Yuen et al. [31] found that lurasidone restored NMDAR-mediated synaptic responses to normal levels in a PCP model of schizophrenia by antagonizing 5HT_7_ receptors. The study by Horisawa et al. [32] also demonstrated that lurasidone can ameliorate MK-801-induced deficits in the passive avoidance test in rats via a 5-HT_7_ receptor antagonistic mechanism. Taken together, these results suggest that the 5-HT_7_ receptor antagonistic activity of lurasidone plays an important role in its pharmacological actions of cognition-improving effects.

Moreover, 5-HT_1A_ partial agonism [33,34,35] and lack of D_4_ receptor blockade [36,37], may also affect cognitive function. Previous studies demonstrated that not only 5-HT_7_ receptor antagonism but also 5-HT_1A_ receptor partial agonism contributed to the ability of lurasidone to ameliorate the subchronic PCP-induced deficit in novel object recognition [34,35]. A marmoset study found that activation of the dopamine D_4_ receptor may improve performance in the object retrieval detour task, whereas concomitant blockade of dopamine D_4_ and D_2_ receptors may counteract the effect. Therefore, the lack of affinity for the dopamine D_4_ receptor by lurasidone may also contribute, at least partly, to its cognitive-enhancing effect [37]. Further studies are needed to verify the involvement of the 5-HT_1A_ receptor and the dopamine D_4_ receptor in the pro-cognitive action of lurasidone.

Additionally, lurasidone is capable of promoting neuronal plasticity [38], modulates epigenetic modification [39], and increases the expression of brain-derived neurotrophic factor in cortical and limbic regions of the brain [40,41], all of which may also contribute to procognitive effects.

Currently, there is still a lack of evidence-based pharmacotherapy of bipolar depression in children and adolescents. Due to the chronic and recurrent course of BD in children and adolescents, long-term treatment is often necessary. Therefore, the safety and tolerability of medications are critical considerations. At present, in addition to quetiapine and lurasidone, the recommended therapeutic drugs also include OFC therapy, lamotrigine, and lithium, but their clinical application is conditional due to their adverse reactions [42,43,44]. Notably, antidepressants are not recommended as first-line choices by almost all guidelines because of their risk of inducing manic episodes. Adherence to lurasidone by bipolar patients was approximately equal to or more favorable than other atypical antipsychotics [45,46]. An observational study comparing hospitalization risk in bipolar patients found that lurasidone use was significantly associated with reduced risk for all-cause psychiatric hospitalization compared to quetiapine [46]. Further, increasing studies have confirmed the safety of lurasidone in adolescents with BD. A 2-year double-blind placebo-controlled study found that, for youth with bipolar depression, lurasidone was generally well tolerated, safe, and effective with relatively low rates of discontinuation due to adverse events, and had minimal effects on weight, metabolic parameters, or prolactin [47].

Previous studies have indicated that lurasidone had a low risk in metabolic profiles following short-term or long-term treatment courses in children and adults [17,48,49,50,51,52,53]. Patients who have previously used other second-generation antipsychotics that have led to weight gain have lost weight after switching to lurasidone [54]. Interestingly, a post-hoc analysis based on a placebo-controlled trial in patients with schizophrenia suggested that lower metabolic risk was significantly associated with the improvement in cognitive performance when treated with lurasidone 160 mg/day [55]. In our study, after 8 weeks of treatment, the change in weight was not significantly different between the two groups, as well as other serum metabolic profiles, including triglyceride level, cholesterol level, and fasting blood glucose level. These findings may be explained by the small sample size and short follow-up period in our study. On the other hand, there was also a statistical difference in body weight between the two groups at baseline. The weight of the subjects in the lurasidone group was greater than that in the quetiapine group, which may influence the drug efficacy. Andrea et al. [56] found that excessive weight gain can affect the effectiveness of antipsychotics in patients with schizophrenia. However, the effects of being overweight on the treatment outcomes in patients with BD remain unclear.

Our research has some major limitations. Our small sample size increases the possibility of false negative findings due to limited power. The model we tested should be re-estimated in a larger population. The baseline body weight in the two groups was different, and the patients in the lurasidone group had a higher weight, which may have an impact on the treatment outcome. Longitudinal studies are needed to confirm the impacts of lurasidone and quetiapine on metabolic profiles, symptoms, and cognitive function over a long period of time. Therefore, the findings from our study should be interpreted cautiously and verified in future studies.

## 4. Methods and Materials

### 4.1. Participants

This study enrolled patients aged 10 to 17 years old with BD, based on *DSM-5* criteria, who were experiencing a major depressive episode, from the clinic and wards of the Department of Psychiatry, First Affiliated Hospital Zhejiang University School of Medicine. The 17-item Hamilton Depression Scale (HAMD-17) score of ≥17 points and a Young Mania Rating Scale (YMRS) score of ≤12 were required for study entry. All participants were required to be drug-free from any medication within 8 weeks before enrollment. The Clinical Research Ethics Committee of the First Affiliated Hospital of Zhejiang University approved the study (reference number: IIT20210291B-R1) and written informed consent was obtained from all subjects prior to participation. 

Patients were excluded if they met *DSM-5* criteria for a current or lifetime diagnosis of schizophrenia or any psychotic disorder, substance use disorder, or intellectual disability. Other criteria for exclusion included (1) a history of loss of consciousness caused by organic brain diseases or head trauma, as well as any major or unstable cardiovascular, respiratory, nervous system (including epilepsy or obvious cerebrovascular disease), kidney, liver, endocrine or relevant medical history of immune diseases; (2) clinically significant laboratory indicators (blood, urine routine, and blood biochemical tests) or electrocardiogram abnormalities; (3) suffering from any diseases that may change the absorption, metabolism, or excretion of the study drug; (4) those with hearing problems, who cannot hear or understand the examiner’s speech under normal conversation; (5) the presence of serious suicidal ideation and patients who received electroconvulsive therapy treatment within the last 3 months; (6) those who had previously used study drugs, or have contraindications to study drugs. 

### 4.2. Study Design

In this randomized, blind, and controlled study, eligible patients who met the criteria were randomly assigned, in a 1:1 ratio, into the lurasidone group or the quetiapine group. All the participants then underwent an 8-week intervention study. Study visits were set at screening, week 0 (baseline for randomization), and the end of week 2, 4, and 8, while the cognitive function was assessed at baseline and week 8, respectively. 

The initial therapeutic dose of the lurasidone group was 20 mg per day and gradually titrated up to 60 mg per day within one week. The dose could be increased up to 80–120 mg per day in the next few weeks if necessary. The initial dose of the quetiapine group was 100 mg per day and gradually increased to 300 mg per day within one week. The dose was also flexible and could be increased to 400–600 mg per day in the next few weeks if needed. 

### 4.3. Concomitant Medications

During the 8-week treatment, other antipsychotics, antidepressants, anticonvulsants, and mood stabilizers were not allowed. On an as-needed basis, alprazolam, zolpidem, and zopiclone were allowed for hyposomnia treatment in the first week. Drugs such as propranolol can be used to counteract the extrapyramidal or cardiovascular side effects of drugs. 

### 4.4. Assessment of Cognitive Function 

The primary outcome of the study was set as the difference in cognitive function at the end of week 8 between the lurasidone and quetiapine groups. The cognitive function of the subjects was evaluated using the THINC Integrated Tool, known as THINC-it. The THINC-it instrument is administered via computer or tablet and comprises variations of carefully selected, well-known cognitive assessments: the Choice Reaction Time (CRT) paradigm (i.e., THINC-it: Spotter), which is used to test the subjects’ reflexes; the One-Back Test (OBK) (i.e., THINC-it: Symbol Check), which examines response time and accuracy and can be used to assess subjects’ working memory, executive function, and attention; the Digit Symbol Substitution Test (DSST; i.e., THINC-it: Code breaker), which was used to test participants’ ability to observe, react and execute; and the Trail Making Test-Part B (TMT-B) (i.e., THINC-it: Trails), which was used to assess subjects’ executive function; supplemented by the subjective, self-reported Perceived Deficits Questionnaire for Depression–5-item (i.e., THINC-it: PDQ-5-D). The THINC-it instrument provides an easy and immediate summary of specific test results, which can be completed by patients in approximately 10 to 15 min with minimal instruction, and patient performance results are immediately available [57].

### 4.5. Assessment of Depressive Levels

The secondary outcome was the improvement of depression symptoms at the end of week 2, 4, and 8. HAMD-17 total score was used to assess the depression severity at each time point. The secondary outcome also included the response rate and the remission rate at each evaluation point. Treatment response was defined as at least a 50% reduction in depression severity from baseline to each follow-up visit on validated rating scales. Remission was defined by the following composite criteria: the HAMD-17 total score ≤ 7 and the YMRS total score ≤ 8.

### 4.6. Assessment of Weight and Serum Metabolic Profiles

In addition, the rate of change in weight and serum metabolic profiles (triglyceride, cholesterol, and fasting blood glucose) were also designated as the secondary outcomes.

### 4.7. Safety and Tolerability

Safety and tolerability assessments included the incidence and severity of adverse events (AE), which were reviewed and recorded during each follow-up visit. Physical examination and vital signs were collected at each interview, laboratory tests (including blood routine, biochemical, fasting insulin, and thyroid function tests) and a 12-lead electrocardiogram were collected at the end of week 0, 4, and 8.

### 4.8. Statistical Analysis

In this study, the intention-to-treat (ITT) method was used to analyze the main outcomes. ITT analysis includes all randomized participants in the groups to which they are randomly assigned, regardless of their adherence to the entry criteria, the treatment they actually receive, and subsequent withdrawal from treatment or deviation from the protocol [58].

The distribution characteristics of data were first confirmed. For normally distributed data, mean and standard deviation (SD) were used to describe the central tendency and dispersion of the data, while for non-normally distributed data, the median (M), the 25% percentile (P25), and the 75% percentile (P75) were used. Continuous variables were compared using the T-test or Mann–Whitney U test, and categorical variables were compared using the chi-square test (χ^2^) test or Fisher’s exact method and expressed as frequency (percentage). Baseline demographic and clinical variables were compared using the aforementioned methods. Covariance analysis was used to compare the differences in cognitive function between the two groups, using baseline data as a co-variable. In addition, regarding missing values and correlation among repeated measurements, we used generalized estimation equations (GEE) to compare the depression score, the response rate, the remission rate, and the rate of change in metabolic parameters.

All statistical tests were performed using SPSS Version 25.0 (IBM Corp., Armonk, NY, USA). *p* ≤ 0.05 was considered as statistical significance (two sides).

## 5. Conclusions

Taken together, the findings of this study provided preliminary evidence that both lurasidone and quetiapine can effectively improve depressive symptoms in children and adolescents with bipolar depression, and lurasidone may be superior to quetiapine in improving cognitive function. Further studies with a better design are warranted to verify our findings.

## Figures and Tables

**Figure 1 pharmaceuticals-15-01403-f001:**
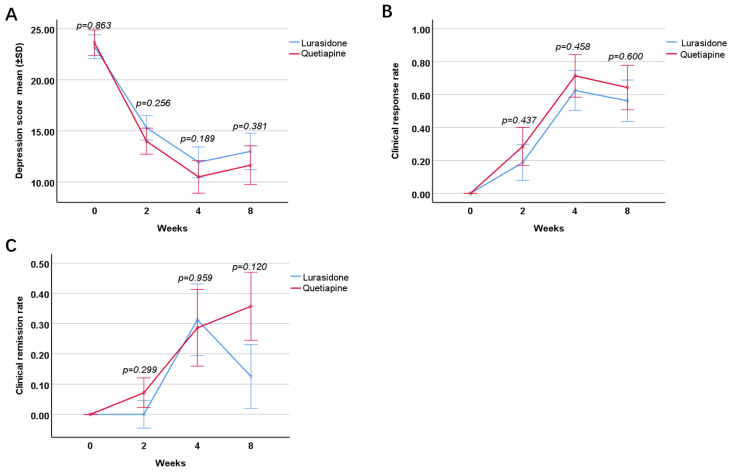
Improvement in depression following quetiapine or lurasidone treatment. Comparison of depression severity by HAMD scores between the two groups at baseline, and the end of week 2, 4 and 8 (**A**); comparison of clinical response rate between the two groups at the end of 2, 4, and 8 (**B**); comparison of clinical remission rate between the two groups at the end of 2, 4 and 8 (**C**).

**Figure 2 pharmaceuticals-15-01403-f002:**
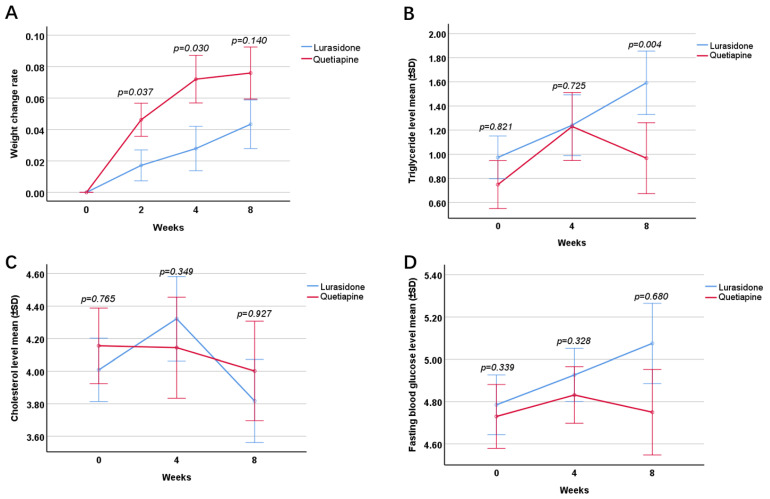
Comparison of metabolic profiles between the two treatment groups. Comparison of the rate of change in body weight between the two groups at the end of 2, 4, and 8 (**A**); comparison of triglyceride levels between the two groups at baseline, 4 and 8 weeks (**B**); comparison of cholesterol levels between the two groups at baseline, 4 and 8 weeks (**C**); comparison of fasting blood glucose levels between the two groups at baseline, 4 and 8 weeks (**D**).

**Table 1 pharmaceuticals-15-01403-t001:** Demographic and clinical characteristics at baseline.

Characteristic	Lurasidone (*n* = 29)	Quetiapine (*n* = 32)	t/χ^2^/ U	*p*
Age M (P_25_, P_75_) (years)	14 (14, 16)	15 (14, 16)	4.45 ^c^	0.035 *
Gender (male/female)	6/23	10/22	0.88 ^b^	0.349
Level of education M (P_25_, P_75_) (years)	8 (7, 10)	10 (8, 10)	4.89 ^c^	0.027 *
Duration of illness M (P_25_, P_75_) (months)	16 (12, 30)	22 (12, 30.75)	0.01 ^c^	0.925
Height mean±SD (cm)	165.07 ± 7.61	164.19 ± 6.59	0.49 ^a^	0.630
Weight M (P_25_, P_75_) (kg)	60 (46, 73)	49.5 (44.88, 57.75)	4.12 ^c^	0.042 *
Smoking(yes/no)	2/27	0/32	0.63 ^b^	0.429
Drinking(yes/no)	3/26	2/30	0.01 ^b^	0.909
Psychotic symptoms (with/without)	12/17	4/28	6.56 ^b^	0.010 *
Family history (positive/recessive)	5/24	4/28	0.03 ^b^	0.873

^a^*t*-test; ^b^ χ^2^ test; ^c^ Mann–Whitney U tests; * statistically significant.

**Table 2 pharmaceuticals-15-01403-t002:** Comparison of the change in cognitive function between the two groups before and after treatment.

Project	Lurasidone (Mean ± SD)	Quetiapine (Mean ± SD)	LSM, 95% CI	t	*p*
Spotter	0.07 ± 0.02	0.07 ± 0.03	0.001, −0.076–0.074	0.03	0.978
Symbol Check reaction time	0.08 ± 0.02	0.01 ± 0.02	0.07, 0.02–0.11	2.84	0.008 *
Symbol Check accuracy	0.20 ± 0.04	0.05 ± 0.04	0.16, 0.04–0.28	2.68	0.012 *
Code breaker	9.19 ± 2.80	11.29 ± 3.00	2.10, −10.62–6.43	0.51	0.618
Trails	6.18 ± 2.01	4.35 ± 2.15	1.83, −4.20–7.86	0.62	0.538
PDQ-5-D	2.35 ± 1.02	5.37 ± 1.13	3.10, −6.25–0.21	1.92	0.066

* Statistically significant.

## Data Availability

Data is contained within the article.

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
