# Peer review of "Lurasidone versus Quetiapine for Cognitive Impairments in Young Patients with Bipolar Depression: A Randomized, Controlled Study"

_pharmaceuticals, 2022, doi:10.3390/ph15111403_

Round 1

Reviewer 1 Report

The authors provide an interesting study comparing clinical efficacy of two antidepressants in young participants with bipolar depression, and found their antidepressive effects are equivalent. The study provide more evidence for understanding the drug treatment of bipolar depression. The manuscript is well written.
Minor comments are:
There is disparity of weight between the groups, with a much higher weight in the lurasidone group, which may impact the drug treatment. The authors may discuss this limitation.

Reviewer 2 Report

Thank you for giving me the opportunity to read and comment a report “Lurasidone versus quetiapine for cognitive impairments in young patients with bipolar depression: a randomized, controlled study” by Diao X. et al.

In the reviewed manuscript, the possible superiority of lurasidone over quetiapine in improving cognitive function in children and adolescents with bipolar depression has been investigated.

This paper is well written, correctly structured with a suitable research concept, the study limitations are addressed, and it is of relevance to readers of the journal.

However, I include a few comments for your consideration.

·  According to the rules of the journal the abstract must be without headings.

·      It would be desirable for the authors to provide more detail on the main aim of the study.

·   Medians, Means and percentiles have been included in the results section, but not described in the statistical analysis subsection.

·     In the statistical analysis subsection, the p value has been set at 0.05. However, it should be determined whether it is less (<) or less than or equal (≤) to 0.05.

·   The term "statistically significant" is used throughout the manuscript. This is not necessary, just indicating the p-value is enough.

·      In Table 1 there are abbreviations that are not defined as M or P. Does the abbreviation s stand for standard deviation? In that case the abbreviation would be SD. Please clarify.

·     It would be convenient to review the format of table 2. When the table was included in the manuscript, the table footnote was misplaced. As in Table 1, authors are encouraged to review the abbreviations to ensure that all abbreviations are defined.
